

# Dynamic gene expression profiles during postnatal development of porcine subcutaneous adipose

Jie Zhang[1,2,*], Jideng Ma[1,*], Keren Long[1,*], Long Jin[1], Yihui Liu[1], Chaowei Zhou[1,3], Shilin Tian[4], Lei Chen[5], Zonggang Luo[2,5], Qianzi Tang[1], An'an Jiang[1], Xun Wang[1], Dawei Wang[4], Zhi Jiang[4], Jinyong Wang[5], Xuewei Li[1] and Mingzhou Li[1]

[1] Institute of Animal Genetics and Breeding, College of Animal Science and Technology, Sichuan Agricultural University, Ya'an, Sichuan, China
[2] Department of Animal Science, Southwest University at Rongchang, Chongqing, China
[3] Department of Aquaculture, Southwest University at Rongchang, Chongqing, China
[4] Novogene Bioinformatics Institute, Beijing, China
[5] Chongqing Academy of Animal Science, Chongqing, China
* These authors contributed equally to this work.

Corresponding authors
Xuewei Li, xuewei.li@sicau.edu.cn
Mingzhou Li,
mingzhou.li@sicau.edu.cn

## ABSTRACT

A better understanding of the control of lipogenesis is of critical importance for both human and animal physiology. This requires a better knowledge of the changes of gene expression during the process of adipose tissue development. Thus, the objective of the current study was to determine the effects of development on subcutaneous adipose tissue gene expression in growing and adult pigs. Here, we present a comprehensive investigation of mRNA transcriptomes in porcine subcutaneous adipose tissue across four developmental stages using digital gene expression profiling. We identified 3,274 differential expressed genes associated with oxidative stress, immune processes, apoptosis, energy metabolism, insulin stimulus, cell cycle, angiogenesis and translation. A set of universally abundant genes (*ATP8*, *COX2*, *COX3*, *ND1, ND2*, *SCD* and *TUBA1B*) was found across all four developmental stages. This set of genes may play important roles in lipogenesis and development. We also identified development-related gene expression patterns that are linked to the different adipose phenotypes. We showed that genes enriched in significantly up-regulated profiles were associated with phosphorylation and angiogenesis. In contrast, genes enriched in significantly down-regulated profiles were related to cell cycle and cytoskeleton organization, suggesting an important role for these biological processes in adipose growth and development. These results provide a resource for studying adipose development and promote the pig as a model organism for researching the development of human obesity, as well as being used in the pig industry.

## INTRODUCTION

Adipose is an anatomical term for loose connective tissue composed of adipocytes (*Poissonnet, LaVelle & Burdi, 1988*). It is recognized as more than an energy reservoir for

lipid storage; in fact, it plays important roles in the pathogenesis of obesity and related comorbidities by secreting cytokines that are involved in the regulation of metabolism (*MacDougald & Burant, 2007*; *Rosen & Spiegelman, 2006*; *Mori et al., 2012*). Adipose, unlike most other tissues/organs, grows and develops continuously throughout the lifespan via differentiation of pre-adipocytes into adipocytes, as well as controlling the storage and mobilization of lipids (*Boucher et al., 2012*; *Lijnen et al., 2006*), and it is involved in extracellular matrix proteolysis, adipogenesis, and angiogenesis (*Crandall, Hausman & Kral, 1997*). The size of the adipose tissue mass is defined by both adipocyte number (i.e. hyperplastic growth) and size (i.e. hypertrophic growth) (*Albright & Stern, 1998*).

Domestic pigs (*Sus scrofa*) are an attractive model for studying energy metabolism and obesity given that they share similar anatomical and physiological characteristics with humans, including the cardiovascular, urinary, integumentary, digestive systems and devoid of brown fat postnatally (*Spurlock & Gabler, 2008*; *Swindle et al., 2012*). Furthermore, both humans and pigs are prone to the development of obesity and related cardiovascular diseases such as hypertension and atherosclerosis (*Koopmans & Schuurman, 2015*). Previous research indicated that rodents and humans have existed mark differences in metabolism and adipose tissue biology (i.e. adipsin, TNFα, resistin), and pig models can fill the gap (*Arner, 2005*). They also provide abundant tissue for analyses or repetitive sampling. Pigs also provide the advantage of highly homogeneous genetic backgrounds, the ability to impose a homogeneous feeding regime, and a relatively short generation interval (12 months).

Here, we present a comprehensive survey of gene expression changes in the porcine Hypodermal Layer of Backfat (HLB) across four postnatal developmental stages (0, 30, 180 days and 7 years after birth, hereafter referred to as birth, 30 d, 180 d, and 7 y, respectively). These four representative time points cover major morphological and physiological changes in pig growth and development, namely the newborn (birth) and post-weaning periods (30 d), the point at which peak commercial value for pork yield (180 d) is reached in the modern pig industry (*Fisher et al., 2013*), through to "middle-aged" (7 y) (*Peters & Johnson, 1997*; *Bhathena, Berlin & Johnson, 1996*). Given that the average life span of a domestic pig tends to be 10–15 years (with each year equivalent to ~5 years in a human), 7 years represents the midpoint of the pig's life span and is therefore the starting point of age-related deterioration of physiological functions and metabolism (*Nair, 2005*; *Pollack et al., 2002*).

We identified genes Differentially Expressed (DE) during the development process and performed functional enrichment analysis for DE genes. We found co-expressed genes that are highly expressed during development, which are therefore potentially marker genes of this process. We also used a Short Time-Series Expression Miner (STEM) method to identify sets of DE genes that are linked to different adipose phenotypes. We envision that this study will serve as a valuable resource in adipose studies, and will promote the pig as a model organism for research into the development of human obesity, as well as being highly relevant for the pig industry.

## MATERIALS AND METHODS

### Ethics statement

All research involving animals was conducted according to the Regulations for the Administration of Affairs Concerning Experimental Animals (Ministry of Science and Technology, China, revised in June 2004) and approved by the Institutional Animal Care and Use Committee in College of Animal Science and Technology, Sichuan Agricultural University, Sichuan, China under permit No. DKY-B20110801. Animals were fed according to the nutrition standards by National Research Council (NRC, 1998) and the feeding conditions (including temperature, humidity etc.) were strictly controlled. All animals were fasted 24 h prior to slaughter and were slaughtered simultaneously, with the exception of newborn animals which were not fasted before slaughter.

### Animals and tissues collection

A total of 12 healthy female Jinhua pigs (a fatty Chinese native breed) were used in this study from four postnatal developmental stages: 0 d (birth), 30 d, 180 d, and 7 y. Each stage included three individuals, which were regarded as biological replicates. Venous blood (50 ml) was collected in pre-chilled tubes from each pig immediately before sacrifice. After sacrifice, the HLB, Abdominal Subcutaneous Adipose (ASA), Retroperitoneal Adipose (RAD), Mesenteric Adipose (MAD), Greater Omentum (GOM), *Longissimus Dorsi* Muscle (LDM), *Psoas* Major Muscle (PMM), Cardiac Muscle (CM), liver, spleen, lung and brain were rapidly separated from each carcass, immediately frozen in liquid nitrogen, and stored at −80 °C until RNA and DNA extraction.

### Measurement of adipose-related phenotype

Measurements of concentrations of 8 serum-circulating indicators of metabolism and adipocyte volume are from our previous report (*Li et al., 2012*). Serum concentrations of Total Cholesterol (TC), Triglycerides (TG), High Density Lipoprotein (HDL), Low Density Lipoprotein (LDL), Very-Low Density Lipoprotein (VLDL), Lipoprotein a (Lip-a), Apolipoprotein A1 (Apo-A1) and Apolipoprotein B (Apo-B) were determined by using CL-8000 clinical chemical analyzer (Shimadzu, Kyoto, Japan) via standard enzymatic procedures. The adipocyte volume were measured using Hematoxylin-Eosin (H&E) staining method. The mean diameter of an adipocyte was calculated as the geometric average of the maximum and minimum diameter, and 100 cells were measured for each sample in randomly selected fields. The mean adipocyte Volume (V) was obtained according to the following formula: $V = \pi/6 \ \Sigma fi \ Di^3/\Sigma fi$, where $Di$ is the mean diameter; $fi$ denotes number of cells with that mean diameter $Di$.

### RNA and DNA extraction

Total RNA was extracted from frozen tissues using TRIzol (Invitrogen, Carlsbad, CA, USA) and further purified using an RNeasy column (Qiagen, Valencia, CA, USA) according to the manufacturer's protocol. RNA integrity and concentration were analysis with the Bioanalyzer 2100 (Agilent Technologies, Santa Clara, CA, USA). DNA was

isolated from frozen tissues using the DNeasy Blood & Tissue Kit (Qiagen) follow the manufacturer's instructions.

## Digital Gene Expression (DGE) library preparation and sequencing

The DGE libraries preparation for HLB was carried out using Illumina Gene Expression Sample Prep Kit according to the manufacturer's protocol. In brief, mRNA was isolated from 4 μg of total RNA by binding the mRNA to a magnetic Oligo dT beads. Double strand cDNA were synthesized using Oligo dT beads. Subsequently, the cDNA samples were digested using the restriction enzyme *Nla*III, which recognizes and cuts the most 3′ "CATG." Once digested into fragments, cDNA were ligated to Illumina adapter 1, which contains a recognition site for enzyme *Mme*I, and the enzyme *Mme*I was used to create the 17 bp tag. The Illumina adapter 2 was ligated after *Mme*I digestion and obtains 21 bp tag derived from a single transcript with Illumina adapters attached to both ends. A 15 cycle PCR was performed with two primers (primer GX1 and primer GX2) that anneal to the ends of the adapters to enrich the adapter-ligated cDNA construct. The amplified cDNA construct was purified from a 6% TBE PAGE gel. During the QC steps, Agilent 2100 Bioanaylzer and ABI StepOnePlus Real-Time PCR System are used in quantification and qualification of the sample library. Finally, massively parallel sequencing by synthesis was performed on the Illumina HiSeq™ 2000. Image recognition and base calling were performed using the Illumina Pipeline.

## Analysis of DGE libraries

The raw data were filtered after data processing. The filtering pipeline is, as follows: 1) empty reads removal (reads with only 3′ adaptor sequences but no tags, 2) low quality tag removal (tags with unknown sequences, N), 3) removal of tags too long or too short, leaving only those 21 nt in length; 4) removal of tags with a copy number of 1 (probably resulting from a sequence error), and 5) generation of clean tags. The raw datasets have been submitted to NCBI Gene Expression Omnibus database with the accession number GSE46755. All possible CAGT+17 nucleotide tags were created by using *sus scrofa* UniGene from Ensembl. All clean tags were mapped to the reference sequences (*Sscrofa* 10.2) and only 1 bp mismatch was allowed. The numbers of mapped clean tags was calculated for each library and were then normalized to Transcripts Per Million tags (TPM). To identify DE genes ($P < 0.01$) for the clustering analysis, we used one-way repeated-measures ANOVA for comparisons. Resulting $P$-values of above tests were corrected with adjusted Bonferroni method (FDR < 0.01, 1,000 permutations).

## Measurement of mitochondrion DNA (mtDNA) content

The relative mtDNA copy number was determined by quantitative PCR (q-PCR) with primers for the mitochondrial ATP synthase 6 (*ATP6*), Cytochrome c Oxidase 1 (*COX1*) and NADH Dehydrogenase 1 (*ND1*) genes. The nuclear encoding gene Glucagon (GCG) was simultaneously used as an endogenous control gene for normalization (see Table S1 for primer sequences). All reactions were performed in triplicates, and negative controls

(without template) were always included. Relative mtDNA copy number per diploid cell = (No. of copies of the mtDNA gene)/(No. of copies of GCG), data are expressed as mean ± SD.

## Analysis of development-related expression patterns

Gene expression patterns during the investigated four growth and development stages were identified using STEM software (*Ernst, Nau & Bar-Joseph, 2005*; *Ernst & Bar-Joseph, 2006*), which applied for the clustering, comparing, and visualizing gene expression data from short time series experiments (in general, ~8 time points or fewer). STEM implements a novel method for clustering short time series expression data that can differentiate between real and random patterns. The expression patterns were judged to be statistically significant between the number of genes expected ($n_{(E)}$) and the number of genes assigned ($n$) and potentially contained genes that were coordinately regulated.

To visualize the correlations between genes within the profiles, we constructed colored heat maps by plotting pair-wise correlation values of expression of all the genes within the profiles was performed with MultiExperiment Viewer (MeV) (*Saeed et al., 2003*). The average gene expression value of each significant cluster profile was correlated with each of nine obesity phenotypic traits using a non-parametric Spearman rank correlation coefficient with Bonferroni correction.

## Gene functional enrichment analysis

A Gene Ontology (GO) and KEGG pathway enrichment analysis was used DAVID (Database for Annotation, Visualization and Integrated Discovery) web server (http://david.abcc.ncifcrf.gov/) (*Huang, Sherman & Lempicki, 2008*). The *P* values (i.e. EASE score), which indicated the significance of the comparison, was calculated by Benjamini-corrected modified Fisher's exact test. Only GO and pathway categories with a *P* value less than 0.05 were considered as significant and listed.

## DE genes in QTLs region

QTL data were downloaded from the Pig Quantitative Trait Locus database (PigQTLdb: http://www.animalgenome.org/QTLdb/pig.html) website (*Hu et al., 2013*). PigQTLdb release 23 (April 21, 2014) contains 10,497 QTLs from 416 publications representing 647 different pig traits. Here, we defined QTL genes as those that have an overlapping region with QTL regions, and the overlapping region is at least half the length of the gene or the QTL region, whichever is shorter. In this study, ~282.57 Mb QTL regions of the 2,311 genes were used for analysis. These were assembled from 901 high confidence and narrowed (<2 Mb) QTL affecting fatness and fat composition.

## q-PCR validation

Total RNA were treated with RNase-free DNase I (TaKaRa, Katsushika, Tokyo, Japan). cDNA synthesis and q-PCR was performed using the SYBR® Prime- Script® RT-PCR Kit (TaKaRa) on a CFX96 Real-Time PCR detection system (Bio-Rad, Hercules, CA, USA). The PCR conditions were 5 min at 42 °C, 10 s at 95 °C, and then 40 cycles of 5 s at 95 °C and 30 s at 65 °C. The primers of 12 genes (*TUBA1B, SCD, COX1, COX2, COX3, ATP8,*

*CYTB*, *ND1*, *ND2*, *ND3*, *ND4*, and *ND5*) designed for q-PCR analyses are listed in Table S1. q-PCR for each RNA sample was performed in triplicate. *β* actin (*ACTB*), TATA box Binding Protein (*TBP*) and Topoisomerase II *β* (*TOP2B*) genes were simultaneously used as an internal gene for normalization. The mRNA expression was quantified using the $2^{-\Delta\Delta Ct}$ method, data are expressed as mean ± SD.

## RESULTS AND DISCUSSION

### Phenotypic measurements

As shown in Fig. 1, the body weight (Mann–Whitney U test, $P < 10^{-6}$) and adipocyte volumes (Student's *t*-test, $P < 10^{-4}$) were significantly different among the four stages. Additionally, measurement of eight representative serum adipose metabolism indicators gave the same ranking (One-way ANOVA, $P < 0.05$, Fig. S1). These phenotypic differences at various stages of HLB imply the existence of intrinsic molecular differences.

### Analysis of DGE profiling libraries

To investigate gene expression changes during development, 12 porcine HLB DGE libraries were constructed using Illumina DGE methods. These DGE libraries generated 3.66 to 6.5 million raw tags for each of the 12 libraries. After filtering, the total number of clean tags per library produced ranged from 3.32 to 6.04 million and the number of distinct clean tags ranged from 141,865 to 270,124 (Table S2). To estimate the quality of the DGE data, the saturation and distribution of clean tag expression was analyzed (Figs. S2–S4).

For tag mapping, one reference tag database that included 22,293 sequences from Ensembl *Sscrofa* 10.2 was preprocessed. We obtained 177,693 total reference tag sequences and 164,561 unambiguous tag sequences. Tolerances were set to allow one mismatch in each alignment to take into account polymorphisms across samples. Among the distinct tags, the number that could be mapped to genes ranged from 54,177 to 69,786 (Table S2). A total of 12,618 genes were identified, with at least one tag in all analyzed samples. The level of gene expression was further analyzed by calculating the number of unambiguous tags for each gene and then normalizing this to the number of TPM. The results showed that the majority of genes transcribed were represented by fewer than 10 copies and only a small proportion of genes were highly expressed (Fig. S5). This result suggests that many genes have housekeeping cellular roles (and therefore low expression) and these may be the main regulatory mRNAs in adipogenesis and basal cellular metabolism (*Guo & Liao, 2000*).

Specifically, a set of genes (*ATP8*: ATP synthase protein 8; *COX2*: Cytochrome c Oxidase 2; *COX3*: Cytochrome c Oxidase 3; *ND1*: NADH Dehydrogenase 1; *ND2*: NADH Dehydrogenase 2; *SCD*: Stearoyl-Coenzyme A Desaturase; *TUBA1B*: Tubulin α1b) shared by the four libraries showed over 100-fold increased expression compared with the average level (total expression of all genes/gene number) in each library, which suggests essential roles for these genes in the growth and development of adipose tissue. Stearoyl-coenzyme A desaturase, which catalyzes the de novo synthesis of monounsaturated from saturated fatty acids, contributes to adipocyte differentiation by

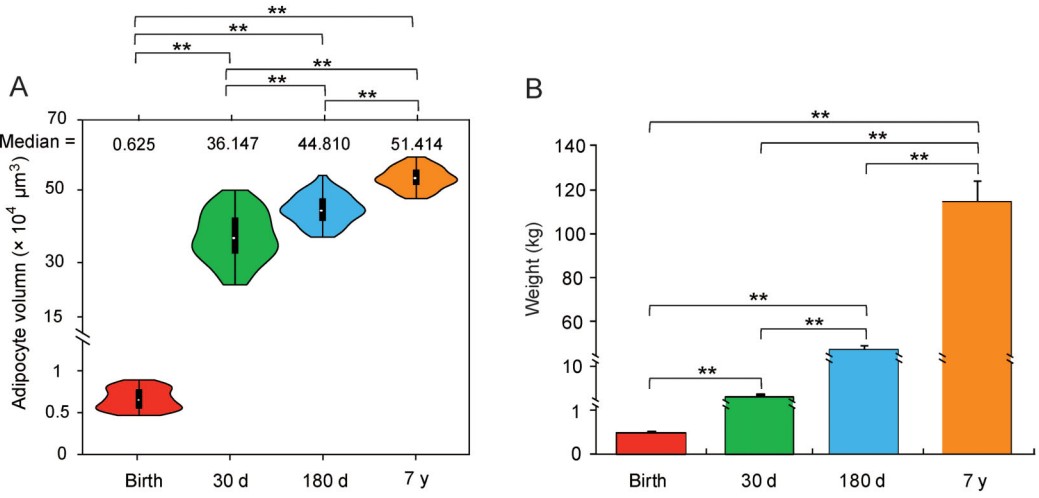

**Figure 1 Differences in phenotype.** (A) Violin plot of adipocyte volume. Each "violin" with the width depicting a 90°-rotated kernel density trace and its reflection. Vertical black boxes denote the Inter-quartile Range (IQR) between the first and third quartiles (25th and 75th percentiles, respectively) and the white point inside denotes the median. Vertical black lines denote the lowest and highest values within 1.5 times IQR from the first and third quartiles, respectively. The statistical significance was calculated by the Mann–Whitney U test (\*\*: $P < 10^{-6}$). (B) The body weight across four stages. The significance of differences among stages was determined by Student's $t$-test (\*\*: $P < 10^{-4}$).

synthesizing monounsaturated fatty acids that are incorporated into the triglycerides of the mature adipocytes (*Dobrzyn & Ntambi, 2005*). Expression of the *SCD* gene increased gradually during porcine adipocyte differentiation, indicating a significant role of *SCD* in this process (*Wang et al., 2004a*). In addition, increased *SCD* gene expression is associated with obesity (*Jones et al., 1996*) and elevated *SCD* gene activity in adipose tissue is closely related to insulin resistance (*Sjögren et al., 2008*). Furthermore, the *SCD* gene has served as a postnatal marker of adipocyte differentiation in bovine subcutaneous adipose tissue (*Martin et al., 1999*). The *TUBA1B* gene has been shown to play an important role in the metabolic actions of insulin, and a mutation in this gene has been associated with insulin resistance (*Kapeller et al., 1995*). Cytochrome c oxidase and NADH dehydrogenase showed increased gene expression levels during adipocyte differentiation and intramuscular fat deposition, suggesting that these genes play important roles in fat development (*Lee et al., 2012*). Deficiencies in *COX1* and *COX2* gene activity have been correlated with a number of human aging diseases such as Alzheimer's and muscle atrophy (*Barrientos et al., 2002*). Obese mice also have higher COX activity levels than lean mice (*Mercer & Trayhurn, 1987*; *Hebert, Ratnasingham & deWaard, 2003*). In addition, the mitochondrial *ATP8* mutation induces mitochondrial Reactive Oxygen Species (ROS) generation, secretory dysfunction, and $\beta$-cell mass adaptation in mice and has been associated with type-2 diabetes and obesity (*Weiss et al., 2012*).

The expression of these genes was validated to ensure the changes were real, and not caused by technical error. q-PCR was applied to investigate the relative expression levels of these genes, and five other genes, in each of the four developmental stages. We correlated

the DGE and q-PCR results, which showed that the expression patterns of these genes were consistent with each other (Pearson's $r > 0.7$, $P < 0.05$; Fig. S6). The fold-change in expression among the stages was often lower when assessed by q-PCR compared with DGE profiling; however, these differences may derive from the intrinsic differences between these two technology platforms (Linsen et al., 2009).

## Analysis of DE genes

To explore global transcriptional changes, we identified 3,274 DE genes across the four developmental stages. Gene ontology (GO) and pathway analysis revealed that DE genes were significantly enriched in the oxidative stress, immune process, apoptosis, energy metabolism process, hormone stimulus, cell cycle, angiogenic and translation categories (Fig. S7), such as 'response to oxidative stress' ($P = 1.74 \times 10^{-4}$, 40 genes), 'regulation of immune system process' ($P = 3.19 \times 10^{-4}$, 52 genes), 'leukocyte activation' ($P = 4.43 \times 10^{-3}$, 48 genes), 'regulation of apoptosis' ($P = 1.94 \times 10^{-2}$, 128 genes), 'phosphorylation' ($P = 1.65 \times 10^{-2}$, 128 genes), 'response to insulin stimulus' ($P = 1.09 \times 10^{-2}$, 23 genes), 'blood vessel development' ($P = 2.21 \times 10^{-2}$, 45 genes) and 'transcription factor binding' ($P = 4.47 \times 10^{-2}$, 83 genes). Oxidative stress accumulating over a lifetime can inflict direct cellular damage and influence various signaling pathways and transcriptional programs regulating key development processes including immunity, energy metabolism, and apoptosis (Finkel & Holbrook, 2000). This suggests that interactions between biological processes affect porcine adipose development and growth. The increased levels of ROS observed in aging adipose tissue are likely a reflection of the modulatory action of age-related levels of hypoxia on adipose function. Hypoxia increases insulin-stimulated glucose uptake and decreases the lipid content of differentiated 3T3-L1 cells (Zhang et al., 2011). In addition, aging-associated oxidative stress reduces the fat mass in mice, while in vitro studies have found that increased oxidative stress through glutathione depletion alters pre-adipocyte differentiation (Zhang et al., 2011). Adipose tissue growth comprises of the enlargement of existing adipocytes and the formation of new cells from committed pre-adipocytes. This is accompanied by the coordinated development of a well-defined vascular system, which ensures that every adipocyte is surrounded by one or more capillaries. Angiogenesis plays an important role in adipose development not only by supplying nutrients and oxygen to nourish adipocytes, but also as a cellular reservoir of adipose precursor and stem cells that control adipose tissue mass and function (Cao, 2013). As part of the immune system, adipose tissue has also been identified to be involved in aging-associated changes in immune and inflammatory responses. Aged visceral fat in mice showed higher levels of the pro-inflammatory cytokines IL-1$\beta$, IL-6, TNF-$\alpha$ and COX-2, but lower expression of the anti-inflammatory PPAR-$\gamma$, chemokine and chemokine receptor (Wu et al., 2007; Lumeng et al., 2011). Starr, Evers & Saito (2009) revealed that IL-6 increased in white adipose tissue with age (Starr, Evers & Saito, 2009). The particular role of adipose tissue in calorie storage makes adipocytes well suited to the regulation of energy balance. Some adipocyte-secreted proteins, including leptin, adiponectin and resistin, play important roles in regulating whole-body metabolism (Galic, Oakhill & Steinberg, 2010;
*Kershaw & Flier, 2004*). To meet the energy requirements of fat and other tissue development, adipocytes allocate energy by responding to hormones, cytokines, and other factors that are involved in energy metabolism (*Frühbeck et al., 2001*; *Klaus, 2004*).

To evaluate the reproducibility of DGE library sequencing, we analyzed the expression variations of the 3,274 DE genes by hierarchical clustering analysis to indicate the reliability of our experimental results as well as operational stability. As shown in Fig. 2A, the three biological replicates were highly correlated with each other (average Pearson's $r = 0.972$, $P < 10^{-16}$, Fig. S8) and all individuals could be clearly assigned to a group, suggesting experimental reliability and further highlighting the low variation in HLB mRNA profiles across different individuals. Interesting, the mRNA expression profiles show an obvious development-specific pattern. Two major branches are defined: one representing birth, 30 d and 180 d, and one representing 7 y. This different clustering pattern may correspond to the intrinsically different biochemical and physiological properties of porcine adipose tissue. Principal components analysis clearly recapitulated these findings. Furthermore, within the treatment groups, gene expression varied significantly by development and time point, as depicted by the well-defined clustering of treatment groups (Fig. 2B). Our previous comparative study of methylation and gene expression profiles of 180 d and 7 y porcine skeletal muscle indicated that in the aged (7 y) pig, symptoms of muscle atrophy have emerged, suggesting that porcine adipose tissue at this point may appear distinct or show specific gene expression patterns compared with earlier stages of growth and development (*Jin et al., 2014*); this also supports the findings of our current study.

We also examined the chromosomal distributions of the 3,274 DE genes, but did not find over- or under-representation of highly DE genes within a particular chromosome, except for the mitochondrion ($P < 10^{-4}$, $\chi^2$-test, Table 1). Through BLAST analysis of the 3,274 DE gene sets against high confidence and narrowed (<2 Mb) QTLs affecting fatness and fat composition in the PigQTLdb (*Hu et al., 2013*), 327 (14.62%) genes were shown to overlap with the defined DE genes (Table 1). This highlights the potential of identifying candidate genes that may be involved in lipogenesis, development and growth. One such gene is the pig Complement Factor B (*CFB*) gene, which was localized to chromosome 7 within the interval of QTLs for backfat thickness, backfat weight, fat area and percentage of backfat in the carcass (*Bidanel et al., 2001*; *Malek et al., 2001*; *Milan et al., 2002*; *Gilbert et al., 2007*). Another gene is Activating Transcription Factor 6 (*ATF6*), which was located in the QTL region for body weight, backfat thickness, backfat weight, and linoleic acid percentage (*Horák et al., 2010*; *Fontanesi et al., 2012*). Meanwhile, obesity is associated with induction of the endoplasmic reticulum stress response signaling and that *ATF6* is the important genes/proteins involved in the signaling regulation (*Agouni et al., 2011*), which supports the notion that pig might be a good model for further adipose studies.

## DE genes involved in inflammatory and lipogenesis process

Numerous evidence indicates that inflammation is a possible underlying basis for the molecular alterations that link aging and age-related pathological processes
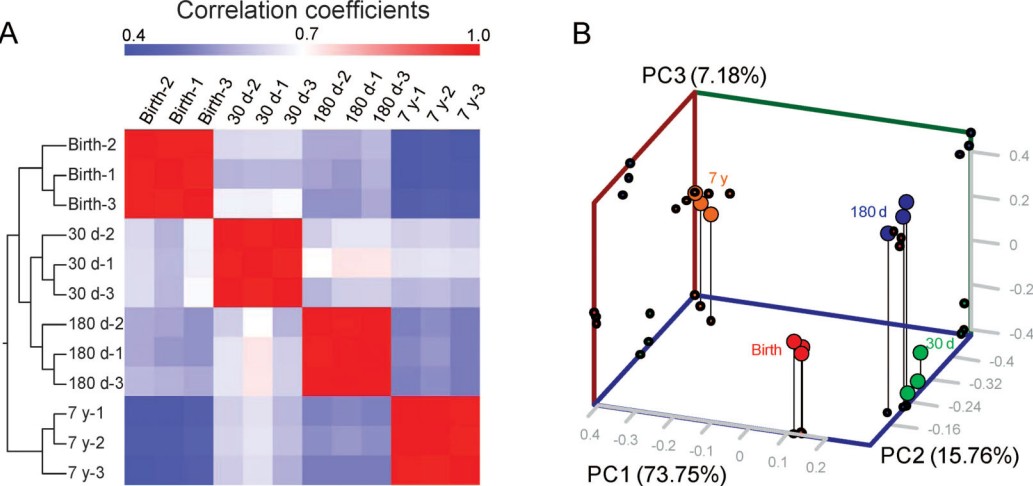

**Figure 2** **Analysis of samples.** (A) Heat map matrix of Pearson's correlation among samples. (B) Three-way PCA plot of samples. The fraction of the variance explained is 73.75%, 15.76% and 7.18% for eigenvector 1, 2 and 3.

**Table 1** DE genes distribution in chromosome and QTLs region.

| Chromosome/ mitochondrion (length, Mb) | Gene number | DE gene number (% of chromosomal/ mitochondrial background) | DE gene number in QTLs region | QTLs region length (Mb) |
|---|---|---|---|---|
| 1 (315.32) | 1,871 | 280 (15.97%) | 28 | 14.98 |
| 2 (162.57) | 1,935 | 301 (15.56%) | 28 | 9.61 |
| 3 (144.79) | 1,289 | 192 (14.9%) | 0 | 0 |
| 4 (143.47) | 1,079 | 178 (16.%) | 55 | 16.02 |
| 5 (111.51) | 1,035 | 153 (14.78%) | 10 | 3.56 |
| 6 (157.77) | 1,700 | 282 (16.59%) | 50 | 16.52 |
| 7 (134.76) | 1,423 | 179 (12.58%) | 38 | 9.47 |
| 8 (148.49) | 707 | 112 (15.84%) | 15 | 4.52 |
| 9 (153.67) | 1,224 | 158 (12.9%) | 12 | 4.11 |
| 10 (79.10) | 426 | 67 (0.23%) | 0 | 0 |
| 11 (87.69) | 352 | 47 (13.35%) | 3 | 2.01 |
| 12 (63.59) | 999 | 182 (18.22%) | 11 | 2.82 |
| 13 (218.64) | 1,298 | 209 (16.1%) | 26 | 11.16 |
| 14 (153.85) | 1,197 | 207 (17.29%) | 18 | 7.06 |
| 15 (157.68) | 809 | 138 (17.06%) | 12 | 3.81 |
| 16 (86.90) | 351 | 53 (15.1%) | 7 | 1.94 |
| 17 (69.70) | 579 | 99 (17.1%) | 4 | 0.46 |
| 18 (61.22) | 434 | 69 (15.9%) | 8 | 3.76 |
| X (144.29) | 733 | 148 (20.19%) | 2 | 0.11 |
| Mitochondrion (0.0167) | 13 | 9** (69.23%) | / | / |

**Notes:**
The statistical significance was calculated by the $\chi^2$-test.
**$P < 10^{-4}$.

(*Chung et al., 2006*). Based on the annotation of the Pathway Central database (SABiosciences, MD, USA), we identified 38 orthologous inflammation-related genes between pig and human, of which only five genes (5/38, 13.15%) were DE across porcine distinct developmental stages, indicating that there was no obvious change in porcine HLB tissue with the aspect of inflammation till middle age (7 y). Nonetheless, the expression level of two inflammatory genes, *CRP* and *IL13RA1*, which plays an important role in anti-inflammatory (*Marnell, Mold & Du Clos, 2005*; *Alfarsi, Hamlet & Ivanovski, 2014*), have been increasing along with development (Fig. S9A), which may serve as early biomarkers of aging-related inflammation. In addition, the expression of other three inflammatory genes, *CX3CR1, CXCL9* and *IL10RA*, showed a steady increase during birth to 180 d followed the sharp decrease in 7 y (Fig. S9B), which also need to pay attention in further aging related studies.

Furthermore, previous studies in human have identified several key candidate genes involved in lipid metabolism, such as *FASN, PPARG, CEBPA, SLC1A4, ADIPOR1* and *ADIPOR2* genes (*Wang et al., 2004b*; *Kersten, Desvergne & Wahli, 2000*; *Yamamoto et al., 2010*). Surprisingly, these genes in our study were not DE across porcine distinct developmental stages in HLB tissue; however, they also showed a trend consistent of their biological function. Lipogenesis genes were high expression in 30 d and 180 d (Fig. S9C), in contrast, lipolysis genes were low expression in 30 d and 180 d (Fig. S9D). Of course, we also found DE genes related lipogenesis, such as *CEBPG, CEBPZ* and *PRKAR1A* genes that had higher expression level in Birth than other stages (Fig. S9E), and *LIPA, HEXB* and *ACE* genes in 180 d had highest expression level across four stages (Fig. S9F), suggesting that the governance of lipogenesis in distinct developmental stage are different genes.

## Analysis of STEM cluster

To explore temporal gene expression patterns, we used the STEM algorithm to search the most probable set of clusters generating the observed time series. The 3,274 DE genes were clustered into 24 distinct expression patterns, of which 1,097 DE genes were significantly over-enriched in six expression patterns (Fig. 3A), which have significantly more genes assigned under the true ordering of four stages than the average number assigned to the model profile in the permutation runs. To investigate whether the change of significantly clustered genes affected phenotype variation, we averaged the expression of each significant cluster profile and performed association analysis with the phenotypic traits (Fig. 3B). We identified that these significant cluster profiles significantly correlated with the amount of Lip-a (Spearman $r = 0.81$, $P = 1.61 \times 10^{-3}$), VLDL (Spearman $r = 0.78$, $P = 2.59 \times 10^{-3}$; Spearman $r = -0.82$, $P = 1.14 \times 10^{-3}$), TG (Spearman $r = 0.83$, $P = 7.85 \times 10^{-4}$; Spearman $r = -0.85$, $P = 5.21 \times 10^{-4}$), LDL (Spearman $r = 0.79$, $P = 1.9 \times 10^{-3}$) and TC (Spearman $r = -0.79$, $P = 2.33 \times 10^{-3}$) in serum.

As shown in Fig. 3C, we found that profiles 1, 2, 3 and 4 are highly positive correlated to each other, as are profiles 5 and 6. We therefore combined these genes for further

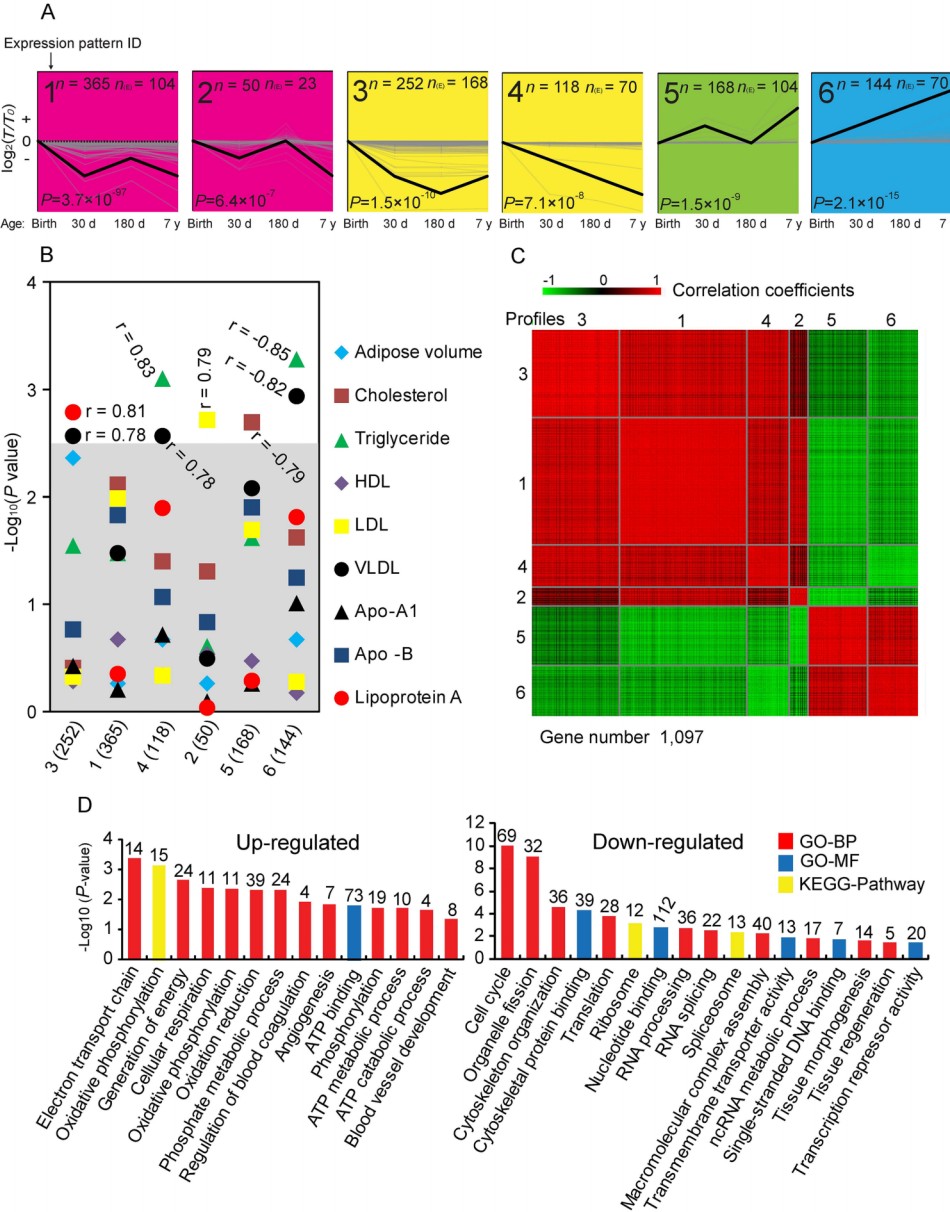

**Figure 3 Analysis of short time-series expression cluster.** (A) Six significant cluster profiles that have significantly more genes assigned under the true ordering of time points compared to the average number assigned to the model profile in the permutation runs (Non-significant cluster profiles are not shown). The upper left represents the serial number of the cluster, the lower left represents the $P$ value (Fisher's exact test), $n$ represents the number of genes assigned, and $n_{(E)}$ represents the number of genes expected. Three other expressions were normalized to the highest one in four stages firstly, and then all expressions were $\log_2$-transformed. (B) Correlations between significant cluster profiles and phenotypic traits. $-$Log $P$ values for Spearman correlation coefficients between the values of the profiles and the different phenotypic traits are shown. The gray shadow represents a highly stringent Bonferroni corrected $P$ value of 0.05. (C) Heat map of significant cluster profiles. Gene pairs strongly positively or negatively correlated are shown in red or green, respectively. (D) Gene ontology (GO) categories enriched for up- and down-regulated genes involve in significant cluster profiles. The $P$ value, which indicated the significance of the comparison, was calculated by Benjamini-corrected modified Fisher's exact test. BP, biological process; MF, molecular function.

functional enrichment analysis. Significant GO categories of up-regulated genes (profile 5 and 6) correlated with phosphorylation and angiogenesis (Fig. 3D). Various well-known genes involved in phosphorylation and angiogenesis of adipose were identified. For example, Pyruvate Dehydrogenase Kinase 2 (*PDK2*) and Pyruvate Carboxylase (*PC*) are involved in phosphorylation, which continually increased with adipose growth and development (Fig. S10), and may play a role in diabetes, lipogenesis and insulin secretion (*Lee, 2014*; *Xu et al., 2008*). Angiopoietin 2 (*ANGPT2*) and Epidermal Growth Factor-Like domain 7 (*EGFL7*) are increased with adipose growth and development in general (Fig. S10), and could be regulating vasculogenesis (*Wang, Huebert & Shah, 2014*; *Bambino et al., 2014*). These results are consistent with these genes and their associated processes having important roles in the process of adipose development. Significant GO categories of down-regulated genes (profiles 1, 2, 3 and 4) correlated with cell cycle and cytoskeleton organization (Fig. 3D), consistent with the knowledge that inhibition of the cell cycle occurs during the later stages of growth and development. Genes involved in cell cycle were identified. For example, minichromosome maintenance complex component 2 and 3 (*MCM2* and *MCM3*), which are involved in the initiation of eukaryotic genome replication (*Kang, Warner & Bell, 2014*); Cyclin D3 (*CCND3*), which functions as a regulator of CDK kinases, whose activity is required for cell cycle G1/S transition (*Li et al., 2013*); F-Box Protein 5 (*FBXO5*), which is a mitotic regulator that interacts with *CDC20* and inhibits the anaphase promoting complex (*Reimann et al., 2001*); both of those genes are decreased with adipose growth and development (Fig. S10).

## DE genes in middle-aged pig

The results of STEM analysis revealed that gene expression patterns are up- or down-regulated in general; however, the gene expression pattern was changed between 180 d and 7 y in profile 3, suggesting that the growth and development of 7 y porcine adipose tissue represents a distinct change. Previous studies mainly carried out in pigs are in neonatal or before they reach the age of 1 year (*Zhang et al., 2013*; *Zhou et al., 2013*). Limited studies have been carried out using young adult pigs (2 years old). Therefore, we used middle-aged pigs (7 years old) as representative of slightly older individuals. A comparative study of methylation and gene expression profiles of 180 d and 7 y porcine skeletal muscles indicated that 7-year-old pigs have emerging symptoms of aging (*Jin et al., 2014*). We therefore investigated whether known age-related genes from the Human Ageing Genomic Resources (HAGR) database (*Tacutu et al., 2013*) were included in the genes found to be DE (7 y vs. birth, 7 y vs. 30 d, 7 y vs. 180 d, $P < 0.01$, Student's $t$-test) in our data set. Within the HAGR, GenAge is a database of 288 genes potentially associated with human age, 61 of which were found to be DE with age in our study ($P = 0.017$, $\chi^2$-test, Table S3). GO analysis revealed that these 61 genes are associated with DNA repair, apoptosis, transcription and immune processes. Various well-known genes were identified, such as Insulin-like Growth Factor 1 (*IGF1*), and Peroxisome Proliferator Activated Receptor α (*PPARα*), a Mechanistic Target of Rapamycin (*MTOR*) and Uncoupling Protein 2 (*UCP2*).

Peer J

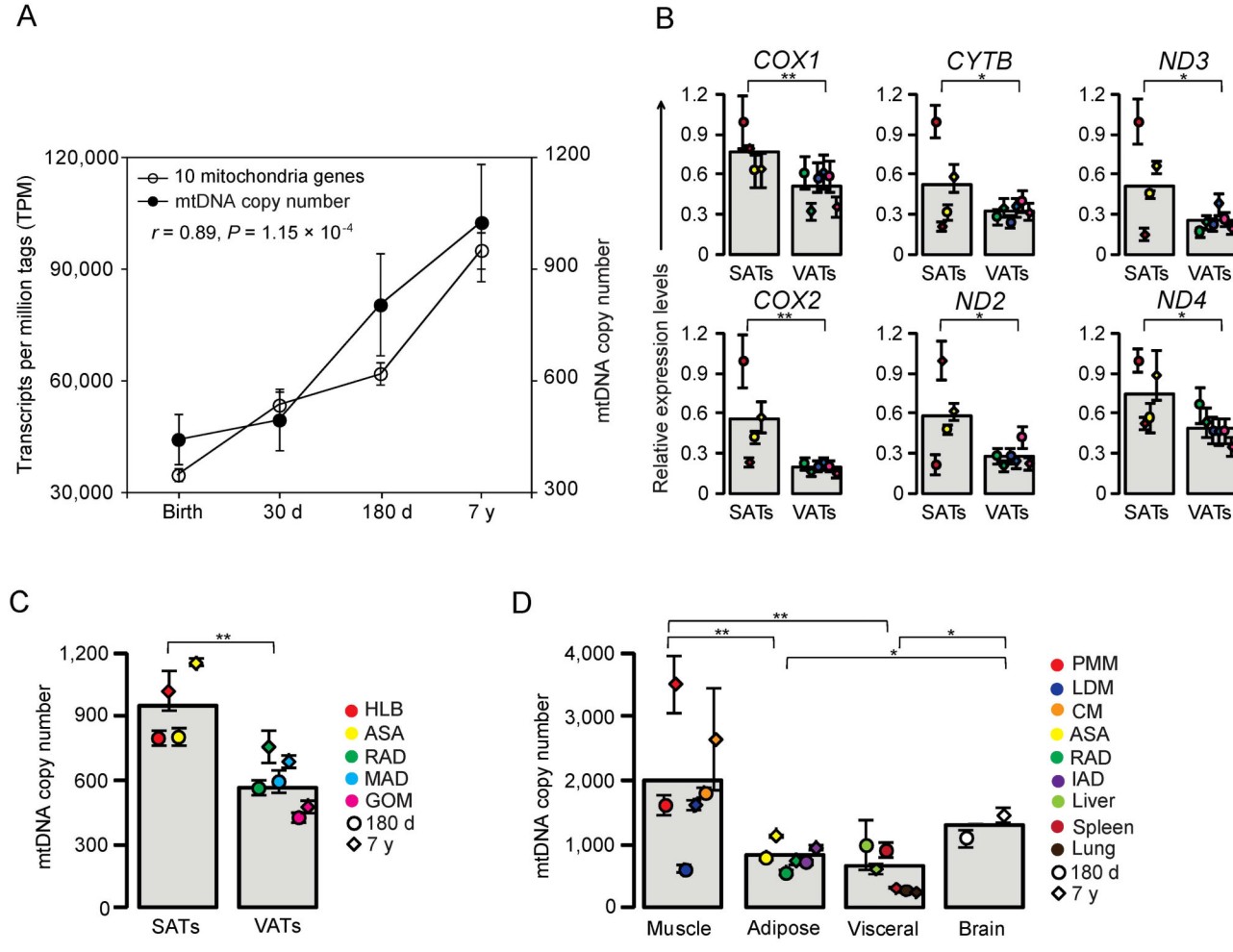

**Figure 4** **Analysis of mitochondrial genes.** (A) Correlation of mitochondria genes expression and DNA copy number. Data are means ± SD. The Pearson correlation coefficient (*r*) and the corresponding significance value (*P*) were shown. (B) Comprehensive survey of development stage- and tissue-specific expression patterns of mitochondrial genes. SAT, subcutaneous adipose tissue; VAT, visceral adipose tissue; *COX1*, cytochrome c oxidase 1; *COX2*, cytochrome c oxidase 2; *CYTB*, cytochrome b; *ND2*, NADH dehydrogenase 2; *ND3*, NADH dehydrogenase 3; *ND4*, NADH dehydrogenase 4. (C) Comprehensive survey of development stage- and tissue-specific mitochondria copy number in various types of adipose tissues. (D) Comprehensive survey of development stage- and tissue-specific mitochondria copy number in various types of tissues. Data are means ± SD. The significance of differences between samples was determined by Student's *t*-test (*: $P < 0.05$; **: $P < 0.01$).

## Highly expressed DE genes were related to mitochondrial energy metabolism

Surprisingly, five of the highly expressed DE genes were mitochondrial genes. It is well known that energy metabolism is closely related to mitochondrial gene expression (*Patti & Corvera, 2010*). To verify that the mitochondrial genes were not in different stages of DE by chance, we performed a $\chi^2$-test and found that out of 13 mitochondrial genes, nine (69.23% $P < 0.05$) exhibited dynamic expression changes during the postnatal development stages in HLB (Table 1). Moreover, a previous study reported that the mtDNA copy number was positively correlated with energy metabolism (*Tang et al., 2000*). We therefore used the mtDNA copy number of HLB as a means of partially

reflecting the energy metabolism. The mtDNA copy number of HLB was significantly increased during the process of development (One-way ANOVA, $P = 9.24 \times 10^{-4}$), which was consistent with the expression of mitochondrial genes in our study (Pearson's $r = 0.89$, $P = 1.15 \times 10^{-4}$, Fig. 4A).

We also quantified the expression level of six mitochondrial genes in various adipose tissues between 180 d and 7 y. Expression of all six genes was higher in SATs than in VATs ($P < 0.05$, Student's $t$-test, Fig. 4B), which was comparable with the relative levels of mtDNA copy number between the two types of adipose tissue ($P < 0.01$, Student's $t$-test, Fig. 4C). These results revealed distinct functional properties of SATs and VATs, in which SATs mainly store energy but VATs are involved with inflammatory and immune processes (*Ibrahim, 2010*). A comprehensive survey of mtDNA copy number across different tissues including brain and muscle was performed. Of all the tissues examined, muscle showed the highest mtDNA copy number (Fig. 4D), which is consistent with the higher energy demand of muscle due to its motor function. The mtDNA copy number of brain was relatively lower than that of muscle, but higher than all the visceral organs and adipose tissues ($P < 0.05$, Student's $t$-test, Fig. 4D), suggesting that muscle and brain have the highest energy requirements (*Belanger, Allaman & Magistretti, 2011*; *Magistretti & Allaman, 2013*).

## CONCLUSION

This study reports a comprehensive transcriptome survey of HLB in four developmental stages, based on directly sequenced gene expression. The data set and research here shed new light on gene regulation during adipose tissue development. Most of the genes highly expressed during development were mitochondrial-related genes, suggesting the alteration of oxidative metabolism during development. Pigs are considered a good biomedical model for human developmental studies because they share the same general physiology. Indeed, the candidate age-related genes found in our study included approximately 20% of the known human age-related genes. Furthermore, of all the candidate genes, some were primarily associated with adipose-related phenotypes, such as triglyceride and cholesterol metabolism, which suggests these genes may serve as potential biomedical markers for mammalian adipose tissue development.

## ACKNOWLEDGEMENTS

We thank Yanmei Xie, Hongmei Wang and Lu Bai for help with experiments. We are grateful to Lingjin Xian, Qingzhi Li and Zhiping Mu for assistance with sample collections.

### Funding

This work was supported by grants from the National Special Foundation for Transgenic Species of China (2014ZX0800950B and 2011ZX08006-003), the National Natural Science

Foundation of China (31472081 and 31402046), the Fund for Distinguished Young Scientists of Sichuan Province (2013JQ0013), the Program for Innovative Research Team of Sichuan Province (15CXTD0048), the Chongqing Agriculture Development Grant (13410), the Program for Changjiang Scholars and Innovative Research Team in University (IRT13083), and the National Biological Breeding Capacity Building and Industrialization Projects Sponsored by National Development and Reform Commission [(2014)2573]. The funders had no role in study design, data collection and analysis, decision to publish, or preparation of the manuscript.

## Grant Disclosures

The following grant information was disclosed by the authors:
National Special Foundation for Transgenic Species of China: 2014ZX0800950B and 2011ZX08006-003.
National Natural Science Foundation of China: 31472081 and 31402046.
Fund for Distinguished Young Scientists of Sichuan Province: 2013JQ0013.
Program for Innovative Research Team of Sichuan Province: 15CXTD0048.
Chongqing Agriculture Development Grant: 13410.
Program for Changjiang Scholars and Innovative Research Team in University: IRT13083.
The National Biological Breeding Capacity Building and Industrialization Projects Sponsored by National Development and Reform Commission: (2014)2573.

## Competing Interests

The authors declare that they have no competing interests.

## Author Contributions

- Jie Zhang performed the experiments, analyzed the data, wrote the paper, prepared figures and/or tables, reviewed drafts of the paper.
- Jideng Ma performed the experiments, analyzed the data, wrote the paper, prepared figures and/or tables, reviewed drafts of the paper.
- Keren Long analyzed the data, prepared figures and/or tables, reviewed drafts of the paper.
- Long Jin analyzed the data, reviewed drafts of the paper.
- Yihui Liu contributed reagents/materials/analysis tools, reviewed drafts of the paper.
- Chaowei Zhou performed the experiments, reviewed drafts of the paper.
- Shilin Tian performed the experiments, reviewed drafts of the paper.
- Lei Chen performed the experiments, reviewed drafts of the paper.
- Zonggang Luo performed the experiments, reviewed drafts of the paper.
- Qianzi Tang performed the experiments, reviewed drafts of the paper.
- An'an Jiang contributed reagents/materials/analysis tools, reviewed drafts of the paper.
- Xun Wang performed the experiments, reviewed drafts of the paper.
- Dawei Wang contributed reagents/materials/analysis tools, reviewed drafts of the paper.
- Zhi Jiang contributed reagents/materials/analysis tools, reviewed drafts of the paper.

- Jinyong Wang contributed reagents/materials/analysis tools, reviewed drafts of the paper.
- Xuewei Li conceived and designed the experiments, reviewed drafts of the paper.
- Mingzhou Li conceived and designed the experiments, reviewed drafts of the paper.

## Animal Ethics

The following information was supplied relating to ethical approvals (i.e., approving body and any reference numbers):

All research involving animals was conducted according to the Regulations for the Administration of Affairs Concerning Experimental Animals (Ministry of Science and Technology, China, revised in June 2004) and approved by the Institutional Animal Care and Use Committee in College of Animal Science and Technology, Sichuan Agricultural University, Sichuan, China under permit No. DKY-B20110801. Animals were allowed free access to food and water under normal conditions, and were humanely sacrificed as necessary, to ameliorate suffering.

## Data Deposition

Genbank: GSE46755.

## Supplemental Information

Supplemental information for this article can be found online at http://dx.doi.org/10.7717/peerj.1768#supplemental-information.

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
