# Peer review of "Dynamic gene expression profiles during postnatal development of porcine subcutaneous adipose"

_PeerJ, doi:10.7717/peerj.1768_

## Round 0.1 · original submission · Minor Revisions

The manuscript is written very well. Experiments were designed and executed carefully. The work is suitable to publish in PeerJ with minor revision.

Reviewer 1 ·

Basic reporting

No Comments

Experimental design

No Comments

Validity of the findings

No Comments

Additional comments

This interesting work, presented by Jie Zhang and coworkers, provides important insights into the development of subcutaneous fat. There is increasing interest in adipose tissue, and the notion that fat depots act as major contributors for a variety of diseases, including (but not limited to) impaired glucose tolerance, diabetes and cancer. The authors are perfectly right when saying that better understanding of adipose tissue “deposition” (I would rather prefer the term “development”) is of critical importance for physiology (and for physiopathology, as well). With the increased prevalence of adiposity in children of the more economically developed countries, the importance of this better knowledge has become primordial. Studies endorsing the concept that an equilibrated adipose tissue is important for health are much appreciated. But, first, it is important to recognize what that means. Thus, the work of the authors in this manuscript is timely and provides an initial “ranging shot” at the expression of genes that are involved in the right/normal development of this tissue.

Overall, the manuscript is well written and fairly easy to follow. The experimental design and methods are well performed, clearly presented and appropriately analyzed. The figures are well also clear and informative. The topic is important and novel data is presented. I have just some minor comments and suggestions that the authors may consider to boost the scientific and translational value of this work.

Discretionary revision:

I work in biomedical research (endocrinology and nutrition) and my area of interest involves the physiology and physiopathology of adipose tissue. I am not sure whether my colleagues and I are the main target of this manuscript. Nevertheless, there are some additional validations that could boost the clinical and translational value of this work:

1) Validation of some candidates (other than mitochondrial genes) in omental adipose tissue (omental fat is believed to be more critical for metabolical disease)
2) Profiling and validation of gene candidates known to be related to obesity and adipose development in humans
3) Provide data from older specimens (insights in the senescence of adipose tissue?). In this respect, I would suggest stress in the GO analysis that revealed increased expression of genes that are related to inflammation and senescence in 7 years-old specimens.
4) It would be interesting highlight and/or assess specific information about some adipocyte-secreted proteins, including interleukin-6, leptin, adiponectin and resistin, common adipo/lipogenic effectors (FASN, PPARg, CEBPa, SREBP, etc.), and genes related with insulin sensitibity (IRS1, SLC1A4, INS1R, etc.), and, thus, playing very important roles in metabolism, at least, in men.
5) I understand that this may be considered beyond the scope of the current work but, trying to endorsing the concept that similar results should be expected in humans (“these results provide a resource for studying adipose development and promote the pig as a model organism for researching the development of human obesity”), it would be grateful some kind of table or summary of results showing how similar pigs and men are, in terms of adipose tissue gene expression, fat distribution and circulating profiles.

Minor Revisions:

Abstract: I would suggest delete the last sentence (somehow overstated). “We identified 3,274 differential expressed genes associated with oxidative stress, immune processes, apoptosis, energy metabolism, insulin stimulus, cell cycle, angiogenesis and translation (P < 0.05)”: differential expressed genes or transcripts? Avoid the p-value or provide the adjusted p-value for each pathway instead. I wouldn’t talk of adipose deposition (maybe fat, lipid or fatty acid synthesis or deposition? Lipogenesis? Circulating fatty acid clearance?).

Reviewer 2 ·

Basic reporting

No Comments

Experimental design

No Comments

Validity of the findings

No Comments

Additional comments

This is a good manuscript that the authors presents a large data-set of transcriptomics results,comparing the porcine subcutaneous adiposetissue across four developmental stages of postnatal development. The experiments and analyses have been performed very well, and the data could support the conclusions. In addition, few studies carried out in pigs after they reach the age of 2 year, the authors associated the pig’s life span with human,and the middle-aged pigs (7 years old) might share much similar physiological functions and metabolism with human in corresponding years, so this would benefit for understanding the application of pig in medicine. It is the opinion of this reviewer that it should be published.

Reviewer 3 ·

Basic reporting

Zhang et al. report that they did a gene expression profiling during postnatal development of porcine subcutaneous adipose by the age of 0 days, 30 days, 180 days and 7 years old pig. While investigating the mRNA transcriptomes in porcine subcutaneous adipose tissue across four developmental stages, they found 3,274 differential expressed genes associated with oxidative stress, immune processes, apoptosis, energy metabolism, insulin stimulus, cell cycle, angiogenesis and translation. Universal abundant genes ATP8, COX2, COX3, ND1, ND2, SCD and TUBA1B were found across all four developmental stages and according to them might play important roles in adipose deposition and development. They identified development-related gene expression patterns that are linked to the different adipose phenotypes. Genes enriched in significantly up-regulated profiles were associated with phosphorylation and angiogenesis whereas genes enriched in significantly downregulated profiles were related to cell cycle and cytoskeleton organization, suggesting an important role for these biological processes in adipose growth and development. Pig might be a model organism for researching the development of human obesity, as well as being used in the pig industry.
The manuscript is well written in a good English language, but some points have to be addressed that it can be published.
Table 1: “mitochnofrional background” has to be corrected to “mitochondrial background”

Supplementary figure 3: some y-axes are labelled from 0-80 % whereas most of them are labelled from 0-100%, they should all look the same, just add the 100% labelling as well. And in the 3nd row in the middle, the dashed line has no explanation, please add it.
Supplementary figure 9: left figures have a y-axis labelling whereas it is missing on all right figures. This should be added.
Q-PCR should be written as qPCR in the whole manuscript as this is the commonly used term for it.
Line 82: should be “were” instead of “are” as experiments are already finished.

Experimental design

What is the criterion that a gene is called differentially expressed? Which fold change or lower bound confidence interval or similar basis for calculation as stringency criterion was used?
Line 92: 2x2Δct method – could you please provide the source for this calculation method? I never heard about it.
Line 190: 2Δct method – also please provide the source for this method. Commonly when going for qPCR the 2-ΔΔct method ((Livak and Schmittgen, 2001) is used.
qPCR validation: according to MIQE standards, PCR conditions should also be mentioned.

Validity of the findings

Data are valid and sound and pig might be a good model for further adipose studies.
But the following might be added:
ATF6 (line 326) is a gene/protein involved in UPR (unfolded protein response) / ER stress regulation. Authors might mention that is it in the meantime obesity is associated with induction of the ER stress signaling and that ATF6 is one on the important genes/proteins involved in the signaling regulation (e.g. Pubmed-ID 21605081).

Additional comments

It is a very nice and interesting manuscript.

---

## Round 0.2 · accepted · Accept

The present state of the manuscript is now suitable for publication in PeerJ.

Reviewer 3 ·

Basic reporting

Line 106/107 (pdf-file): should be “The adipocyte volumes were…”

Experimental design

no comments

Validity of the findings

no comments

Additional comments

Data are valid and sound and pig might be a good model for further adipose studies.